# REYNOLDSFLOW: SPATIOTEMPORAL FLOW REPRESENTATIONS FOR VIDEO LEARNING

## ABSTRACT

Representation learning for videos has largely relied on spatiotemporal modules embedded in deep architectures, which, while effective, often require heavy computation and heuristic design. Existing approaches, such as 3D convolutional modules or optical flow networks, may also overlook changes in illumination, scale variations, and structural deformations in video sequences. To address these challenges, we propose ReynoldsFlow, a physics-inspired flow representation that leverages the Helmholtz decomposition and the Reynolds transport theorem to derive principled spatiotemporal features directly from video data. Unlike classical optical flow, ReynoldsFlow captures both divergence-free and curl-free components under more general assumptions, enabling robustness to photometric variation while preserving intrinsic structure. Beyond its theoretical grounding, ReynoldsFlow remains lightweight and adaptable, combining frame intensity with flow magnitude to yield texture-preserving and dynamics-aware representations that substantially enhance tiny object detection. Experiments on benchmarks with various target scales demonstrate that ReynoldsFlow is consistently comparable to or outperforms existing flow-based features, while also improving interpretability and efficiency. These results position ReynoldsFlow as a compelling representation for video understanding and a strong foundation for downstream model learning. The code will be made publicly available.

## 1 INTRODUCTION

Computer vision techniques are now deeply embedded in everyday life, with video functionality serving as a core feature of modern smartphones. Beyond casual use, video understanding underpins a broad spectrum of applications, including stabilization, interpolation, object detection (Jiao et al., 2021; Zhu et al., 2018a; 2020; 2017), multi-object tracking (Ciaparrone et al., 2020; Agbinya & Rees, 1999; Chen, 2025), and pose estimation (Girdhar et al., 2018; Charles et al., 2016; Von Marcard et al., 2016; Pavllo et al., 2019). Progress in these areas has been largely driven by deep learning architectures such as recurrent neural networks (RNNs) (Zhao et al., 2017; Ebrahimi Kahou et al., 2015), Long Short-Term Memory (LSTM) networks (Graves, 2012; Zhang et al., 2016), and, more recently, transformer- and Mamba-based spatiotemporal models (Tran et al., 2024; Hsu et al., 2023; Gai et al., 2023; Zhang et al., 2021; Li et al., 2024; Park et al., 2024). Despite their strong empirical performance, these methods face notable limitations: they often incur high computational costs, rely on carefully tuned architectural heuristics, and offer limited interpretability, as their representations are rarely grounded in the physical principles of motion.

To address these challenges, we introduce ReynoldsFlow, a physics-inspired representation for video learning. Building on optical flow estimation (Sun et al., 2010), Helmholtz decomposition (Arfken et al., 2011), and the Reynolds transport theorem (RTT) (White, 2011), ReynoldsFlow provides an interpretable analogue to modern spatiotemporal modules, bridging the gap between data-driven architectures and physically principled modeling. By directly deriving video representations from data, ReynoldsFlow reduces dependence on heuristic design, improves computational efficiency, and produces more robust and generalizable spatiotemporal features. Importantly, it operates in a training-free, unsupervised manner, avoiding the need for large labeled datasets or costly optimization. Our main contributions are summarized as follows:

1. **ReynoldsFlow representation**. We introduce ReynoldsFlow, an unsupervised spatiotemporal flow representation derived from the Helmholtz decomposition and the RTT. ReynoldsFlow jointly captures divergence-free and curl-free components, providing a physically interpretable motion descriptor under general assumptions.

2. **Efficiency and plug-and-play adaptability**. We demonstrate that ReynoldsFlow achieves accuracy comparable to SOTA spatiotemporal embedding models while being significantly more computationally efficient. Furthermore, it can be directly computed from raw video data without retraining or task-specific preprocessing, making it a lightweight plug-and-play representation for downstream neural networks.

3. **Comprehensive validation**. We conduct extensive experiments across multiple video datasets, showing that ReynoldsFlow is both highly interpretable and effective in enhancing diverse tasks, including pose estimation, action recognition, and object detection.

## 2 RELATED WORK

### 2.1 OPTICAL FLOW AND VIDEO MOTION ESTIMATION

Early optical flow (OF) methods such as Horn-Schunck (Horn & Schunck, 1981) and Lucas–Kanade (Lucas & Kanade, 1981) laid the foundation for motion estimation. Horn–Schunck enforced global smoothness constraints but struggled with large displacements and motion boundaries. In contrast, Lucas–Kanade offered computational efficiency but was limited to small, consistent motions and performed poorly in complex or textured scenes. Subsequent refinements, including Optimal Filter Estimation (Sharmin & Brad, 2012) and hybrid methods combining Horn–Schunck and Lucas–Kanade (Bruhn et al., 2005), aimed to balance adaptability and accuracy but remained sensitive to occlusions, noise, and background texture. Advancements such as Farneback's polynomial expansion (Farnebäck, 2003) and Brox's high-accuracy variational method (Brox et al., 2004) improved motion estimation at the cost of increased computational demand. TV-L1 (Zach et al., 2007) introduced noise resilience through total variation regularization, while SimpleFlow (Tao et al., 2012) provided a non-iterative approach, trading off accuracy for speed. RLOF (Senst et al., 2012) advanced feature tracking robustness but remained challenged by high-texture regions. To improve efficiency without sacrificing accuracy, methods like DeepFlow (Weinzaepfel et al., 2013) used deep matching, PCAFlow (Wulff & Black, 2015) applied low-dimensional motion bases, and DIS (Kroeger et al., 2016) optimized dense flow for time-critical tasks.

Deep learning reshaped OF, beginning with FlowNet (Dosovitskiy et al., 2015), the first CNN-based model, which improved recognition but struggled with fine-grained motion. FlowNet2 (Ilg et al., 2017) addressed this with a cascaded refinement architecture. EpicFlow (Revaud et al., 2015) combined sparse matching with variational refinement to improve boundary localization. PWC-Net (Sun et al., 2018) introduced a pyramid, warping, and cost-volume framework for practical multiscale motion estimation, while SpyNet (Ranjan & Black, 2017) focused on computational efficiency for real-time use. RAFT (Teed & Deng, 2020) achieved SOTA performance by iteratively refining dense flow fields using learned contextual correlations. Recent extensions such as RPKNet (Morimitsu et al., 2024), SEA-RAFT (Wang et al., 2024), and DPFlow (Morimitsu et al., 2025) further enhanced temporal consistency and scene understanding through recurrent and domain-adaptive mechanisms. Despite improved accuracy, deep learning-based methods remain data-hungry, computationally expensive, and less practical for out-of-the-box deployment.

A task similar to our UAV detection scenario is explored in (Sun et al., 2023), where a convolutional layer emulating the Lucas-Kanade OF is integrated into the YOLO architecture to enhance UAV detection. Similarly, in (Madake et al., 2023), the authors employ Farneback OF to track a golfer's body posture and swing trajectory, followed by handcrafted feature-based classification. Despite their ingenuity, both approaches suffer from the limited informativeness of conventional OF visualizations, making them less effective for fine-grained motion understanding. To our knowledge, no existing work has shown substantial improvements in object detection or pose estimation by directly incorporating OF images into deep neural networks.

## 2.2 DECOMPOSITION-BASED REPRESENTATIONS

The Helmholtz–Hodge decomposition (HHD) provides a principled mathematical framework for separating a vector field into divergence-free, curl-free, and harmonic components (Schwarz, 2006; Bhatia et al., 2012). This formulation has been widely applied in computer vision and graphics for interpreting flow fields and solving boundary value problems. Early work demonstrated applications of discrete HHD to image processing, enabling structural analysis of visual data (Palit et al., 2005). To enhance robustness, convex optimization formulations were later introduced to regularize image flows and stabilize the decomposition process (Yuan et al., 2008; 2009). Beyond flow fields, decomposition has also played an important role in image representation more broadly, ranging from structural organization in large-scale image databases (Guo et al., 1997), to sparse decomposition methods for separating signal components (Fadili et al., 2009), and multi-modality image fusion based on decomposition and sparse coding (Zhu et al., 2018b).

More recently, decomposition-based techniques have been extended to practical vision tasks, such as illumination compensation and texture enhancement in imaging (Wu et al., 2020). Physics-inspired neural architectures further push this line of research, as in HDNet (Qi et al., 2024), which leverages Helmholtz decomposition to design interpretable deep networks for flow estimation. Collectively, these works demonstrate that decomposition-based representations can improve interpretability and robustness in visual analysis. However, most approaches remain confined to spatial formulations, focusing on single-frame decomposition of images or vector fields while neglecting temporal dynamics. This lack of spatiotemporal integration limits their applicability to modern video understanding tasks, where temporal coherence is critical.

## 2.3 SPATIOTEMPORAL MODULES IN VIDEO LEARNING

Modeling temporal dynamics is essential for effective video understanding, as motion information provides critical cues for object behavior, scene changes, and activity recognition. Early deep learning approaches extended convolutional networks to the spatiotemporal domain using 3D convolutions and recurrent modules to capture temporal dependencies across frames (Xie et al., 2017; Qiu et al., 2017; Peng et al., 2021; Ul Amin et al., 2024). Such spatiotemporal feature learning has been applied to diverse tasks, including semantic video segmentation (Qiu et al., 2017), action clustering (Peng et al., 2021), video deraining (Zhang et al., 2022), and anomaly detection (Ul Amin et al., 2024), demonstrating the broad applicability of learned temporal representations. While effective, these approaches typically rely on supervised training with large annotated datasets or costly optimization, limiting their flexibility and interpretability.

Complementary to purely data-driven methods, physics-inspired strategies provide alternative insights into temporal modeling. Specifically, as presented in (Shih et al., 2001), the authors applied the RTT to treat a video clip as a continuous frame flow, enabling theoretically grounded detection of temporal changes such as shot transitions. In contrast, our approach builds on the same underlying principle but integrates it with flow components separated via the HHD to analyze temporal evolution more comprehensively. This combination allows ReynoldsFlow to directly extract interpretable, spatiotemporal video representations from data in an unsupervised, training-free manner, rather than being restricted to a single task. Consequently, our framework bridges the gap between principled physical modeling and modern spatiotemporal feature learning, producing robust representations that can support a wide range of downstream video understanding tasks.

## 3 REYNOLDSFLOW

In this section, we first introduce the necessary preliminaries and then formulate ReynoldsFlow using the Reynolds transport theorem (RTT) (White, 2011), offering a novel physics-inspired perspective on flow estimation. We subsequently present a dedicated visualization scheme to illustrate ReynoldsFlow features. Finally, we evaluate the runtime performance of our method, highlighting its computational efficiency compared to existing flow estimation approaches.

### 3.1 PRELIMINARIES

#### 3.1.1 HELMHOLTZ DECOMPOSITION

Let $\boldsymbol{v} \in C^1(\Omega, \mathbb{R}^2)$ be a continuously differentiable vector field defined on a domain $\Omega$. According to the Helmholtz decomposition theorem (Arfken et al., 2011), $\boldsymbol{v}$ can be uniquely decomposed into the sum of an irrotational (curl-free) component and a solenoidal (divergence-free) component:

$$\boldsymbol{v} = \boldsymbol{v}_d + \boldsymbol{v}_c, \tag{1}$$

where $\boldsymbol{v}_d$ satisfies $\nabla \cdot \boldsymbol{v}_d = 0$ and $\boldsymbol{v}_c$ satifies $\nabla \times \boldsymbol{v}_c = 0$.

#### 3.1.2 REYNOLDS TRANSPORT THEOREM

Consider a smooth scalar function $f = f(\boldsymbol{p}, t)$ defined on a time-dependent domain $\Omega(t)$. A grayscale video can be represented as $f(\boldsymbol{p}(t^n), t^n)$, where $f$ is the intensity at pixel location $\boldsymbol{p}(t^n)$ in the field of view $\Omega(t^n)$ of the camera, and $t^n$ denotes the time at $n$th frame. The RTT (White, 2011) states that:

$$\begin{aligned}
\frac{d}{dt} \int_{\Omega(t)} f \, dA &= \int_{\Omega(t)} \frac{\partial f}{\partial t} \, dA + \int_{\partial\Omega(t)} f(\boldsymbol{v} \cdot \boldsymbol{n}) \, dS \\
&= \int_{\Omega(t)} \frac{\partial f}{\partial t} \, dA + \int_{\Omega(t)} \nabla \cdot (f\boldsymbol{v}) \, dA \\
&= \int_{\Omega(t)} \left( \frac{\partial f}{\partial t} + \nabla f \cdot \boldsymbol{v} + f \nabla \cdot \boldsymbol{v} \right) dA.
\end{aligned} \tag{2}$$

where $\boldsymbol{v} = \boldsymbol{v}(\boldsymbol{p}, t)$ is the velocity vector at $\boldsymbol{p} \in \Omega(t)$ and time $t$. For video footage, if the brightness in the region $\int_{\Omega(t)} f dA$ is constant and the velocity field $\boldsymbol{v}$ is divergence-free (i.e. $\boldsymbol{v} = \boldsymbol{v}_d$) for all time, then equation 2 reduces to:

$$0 = \int_{\Omega(t)} \left( \frac{\partial f}{\partial t} + \nabla f \cdot \boldsymbol{v}_d \right) dA. \tag{3}$$

Assuming the integrand in equation 3 vanishes pointwise and $\boldsymbol{v}_d$ is piecewise constant, equation 3 reduces to the traditional optical flow (OF). We denote the velocity field satisfying equation 2 as $\boldsymbol{v}_r$, referred to as ReynoldsFlow. To clarify its relation with OF, we rename the divergence-free flow $\boldsymbol{v}_d$ as $\boldsymbol{v}_o$, while the curl-free component $\boldsymbol{v}_c$ serves as the complementary flow (CF).

#### 3.1.3 AREA JACOBIAN IN DOMAIN TRANSFORMATION

Consider a region $\Omega(t)$ evolving over time $t$ under the influence of a vector field $\boldsymbol{v}$. Let $\boldsymbol{p}^n = \begin{pmatrix} x^n \\ y^n \end{pmatrix} \in \Omega^n = \Omega(t^n)$ at time $t^n$, with corresponding vector field $\boldsymbol{v}^n = \begin{pmatrix} v_x^n \\ v_y^n \end{pmatrix}$, and assume an uniform time increment $\Delta t = t^{n+1} - t^n$. Using the explicit Euler method, the domain transformation from $\Omega^n$ to $\Omega^{n+1}$ can be approximated by:

$$\boldsymbol{p}^{n+1} \approx \boldsymbol{p}^n + \boldsymbol{v}^n(\boldsymbol{p}^n)\Delta t. \tag{4}$$

From equation 4, the differential wedge product $dx \wedge dy$ at $\boldsymbol{p}^{n+1}$ can be approximated as:

$$\begin{aligned}
dx^{n+1} \wedge dy^{n+1} &\approx (dx^n + dv_x^n \Delta t) \wedge (dy^n + dv_y^n \Delta t) \\
&= dx^n \wedge dy^n + dx^n \wedge dv_y^n \Delta t + dv_x^n \Delta t \wedge dy^n + dv_x^n \Delta t \wedge dv_y^n \Delta t.
\end{aligned}$$

Using the fact that

$$dv_x = \frac{\partial v_x}{\partial x} dx + \frac{\partial v_x}{\partial y} dy, \quad dv_y = \frac{\partial v_y}{\partial x} dx + \frac{\partial v_y}{\partial y} dy,$$

we obtain

$$dx^{n+1} \wedge dy^{n+1} = dx^n \wedge dy^n + \nabla \cdot \boldsymbol{v}^n \, dx^n \wedge dy^n \, \Delta t + \mathcal{O}((\Delta t)^2).$$

For sufficiently small $\Delta t$, this simplifies to:

$$dx^{n+1} \wedge dy^{n+1} \approx (1 + \nabla \cdot \boldsymbol{v}^n \Delta t) \, dx^n \wedge dy^n. \tag{5}$$

As a result, $(1 + \nabla \cdot \boldsymbol{v}^n \Delta t)$ approximates the Jacobian determinant relating area elements $dA^n$ in $\Omega^n$ to $dA^{n+1}$ in $\Omega^{n+1}$. In other words,

$$dA^{n+1} \approx (1 + \nabla \cdot \boldsymbol{v}^n \Delta t) \, dA^n. \tag{6}$$

## 3.2 DERIVATION OF REYNOLDSFLOW

Recall that $\boldsymbol{v}_o$ denote the traditional OF satisfying the brightness constancy assumption:

$$\frac{\partial f}{\partial t} + \nabla f \cdot \boldsymbol{v}_o = 0,$$

where $\boldsymbol{v}_o$ is assumed piecewise constant over local patches $\omega(t)$. This leads to the classical OF:

$$\boldsymbol{v}_o = -(\nabla f)^\dagger \frac{\partial f}{\partial t}, \tag{7}$$

forming the basis of methods such as Lucas-Kanade and Horn-Schunck. To generalize beyond the brightness constancy and divergence-free assumptions, we invoke the RTT on each local patch $\omega(t)$ combined with the Helmholtz decomposition equation 1 to rewrite:

$$\frac{d}{dt} \int_{\omega(t)} f \, dA = \int_{\omega(t)} \left( \frac{\partial f}{\partial t} + \nabla f \cdot \boldsymbol{v}_o + \nabla f \cdot \boldsymbol{v}_c + f \nabla \cdot \boldsymbol{v}_c \right) dA. \tag{8}$$

Applying the explicit Euler method, the left-hand side (LHS) of equation 8 can be approximated as:

$$\text{LHS} = \frac{1}{\Delta t} \left( \int_{\omega^{n+1}} f^{n+1} \, dA^{n+1} - \int_{\omega^n} f^n \, dA^n \right),$$

$$\approx \frac{1}{\Delta t} \int_{\omega^n} \left[ (I + \nabla \cdot \boldsymbol{v}^n \Delta t) f^{n+1} - f^n \right] dA^n, \quad \text{by equation 6}$$

$$= \int_{\omega^n} \left( \frac{f^{n+1} - f^n}{\Delta t} + f^{n+1} \nabla \cdot \boldsymbol{v}^n \right) dA^n.$$

Next, using the Taylor approximation:

$$f^{n+1} - f^n \approx \frac{\partial f^n}{\partial t} \Delta t + \nabla f^n \cdot \begin{pmatrix} \Delta x^n \\ \Delta y^n \end{pmatrix},$$

and the Helmholtz decomposition of the vector field $\boldsymbol{v}$,

$$\text{LHS} \approx \int_{\omega^n} \left( \frac{\partial f^n}{\partial t} + \nabla f^n \cdot \boldsymbol{v}^n + f^{n+1} \nabla \cdot \boldsymbol{v}^n \right) dA^n,$$

$$= \int_{\omega^n} \left( \frac{\partial f^n}{\partial t} + \nabla f^n \cdot (\boldsymbol{v}_c^n + \boldsymbol{v}_o^n) + f^{n+1} \nabla \cdot \boldsymbol{v}_c^n \right) dA^n.$$

Similarly, the right-hand side (RHS) of equation 8 is

$$\text{RHS} = \int_{\omega^n} \left( \frac{\partial f^n}{\partial t} + \nabla f^n \cdot (\boldsymbol{v}_c^n + \boldsymbol{v}_o^n) + f^n \nabla \cdot \boldsymbol{v}_c^n \right) dA^n,$$

Equating both sides and defining $\delta f^n = f^{n+1} - f^n$ gives

$$\int_{\omega^n} \delta f^n \nabla \cdot \boldsymbol{v}_c^n \, dA^n = 0.$$

Integration by parts leads to the variational form:

$$\int_{\partial \omega^n} \delta f^n \boldsymbol{v}_c^n \cdot \boldsymbol{n} \, dS^n - \int_{\omega^n} \nabla \delta f^n \cdot \boldsymbol{v}_c^n \, dA^n = 0. \tag{9}$$

As usual, we can compute the vector field $\boldsymbol{v}_c^n$ from equation 9 by assuming it remains constant within each local window patch $\omega^n$. Specifically, we compute $\boldsymbol{v}_c^n$ on a $3 \times 3$ window patch, denoted as $\omega_{3 \times 3}^n$. To approximate the boundary integral term in equation 9, we apply Simpson's rule:

$$\int_{\partial \omega_{3 \times 3}^n} \delta f^n \boldsymbol{v}_c^n \cdot \boldsymbol{n} \, dS^n \approx \left[ \delta f_{b,x}^n, \delta f_{b,y}^n \right] \cdot \boldsymbol{v}_c^n, \tag{10}$$

where

$$(\delta f_b^n)_x = \frac{1}{3} \begin{bmatrix} 1 & 4 & 1 \\ 0 & 0 & 0 \\ -1 & -4 & -1 \end{bmatrix} * \delta f^n \text{ and } (\delta f_b^n)_y = \frac{1}{3} \begin{bmatrix} -1 & 0 & 1 \\ -4 & 0 & 4 \\ -1 & 0 & 1 \end{bmatrix} * \delta f^n.$$

The domain integral term in equation 9 is approximated as:

$$\int_{\omega_{3x3}^n} \nabla \delta f^n \cdot \boldsymbol{v}_c^n \, dA^n = \int_{\omega^n} [(\nabla \delta f^n)_x, (\nabla \delta f^n)_y] \cdot \boldsymbol{v}_c^n \, dA^n \approx [(\nabla \delta f_\omega^n)_x, (\nabla \delta f_\omega^n)_y] \cdot \boldsymbol{v}_c^n, \quad (11)$$

where

$$(\nabla \delta f_\omega^n)_x = \begin{bmatrix} 1 & 1 & 1 \\ 1 & 1 & 1 \\ 1 & 1 & 1 \end{bmatrix} * \left( \begin{bmatrix} -1 & 0 & 1 \\ -2 & 0 & 2 \\ -1 & 0 & 1 \end{bmatrix} * \delta f^n \right) \text{ and } (\nabla \delta f_\omega^n)_y = \begin{bmatrix} 1 & 1 & 1 \\ 1 & 1 & 1 \\ 1 & 1 & 1 \end{bmatrix} * \left( \begin{bmatrix} -1 & -2 & -1 \\ 0 & 0 & 0 \\ 1 & 2 & 1 \end{bmatrix} * \delta f^n \right).$$

From equation 9, equation 10 and equation 11, we obtain the irrotational flow field:

$$\boldsymbol{v}_c^n = [(\delta f_b^n)_x - (\nabla \delta f_\omega^n)_x, (\delta f_b^n)_y - (\nabla \delta f_\omega^n)_y]^\perp.$$

Recall that the irrotational vector field $\boldsymbol{v}_c^n$ complements the OF field $\boldsymbol{v}_o^n$ derived from the RTT. Notably, the OF component is also naturally canceled out in equation 9, allowing CF to isolate residual non-motion effects such as illumination changes and non-rigid deformation. Hence, we refer to $\boldsymbol{v}_c^n$ as the complementary component of the OF in the sense of Helmholtz decomposition. To ensure smoothness in practice, we define:

$$\boldsymbol{v}_c^n = \begin{bmatrix} G * (-(\delta f_b^n)_y + (\nabla \delta f_\omega^n)_y) \\ G * ((\delta f_b^n)_x - (\nabla \delta f_\omega^n)_x) \end{bmatrix}, \quad (12)$$

where $G$ represents a Gaussian smoothing kernel. Finally, ReynoldsFlow is defined as $\boldsymbol{v}_R^n = \boldsymbol{v}_o^n + \boldsymbol{v}_c^n$, where $\boldsymbol{v}_o^n$ is given by equation 7 and $\boldsymbol{v}_c^n$ by equation 12.

## 3.3 ReynoldsFlow Representation

Flow is commonly visualized using the HSV color space, where motion magnitude and direction are mapped to hue and saturation (Baker et al., 2011). However, the HSV-to-RGB transformation is highly nonlinear, often introducing perceptual inconsistencies, particularly in low-texture regions, complex illumination, or dynamic motion scenarios. Such limitations can hinder downstream tasks, including tiny object detection, and reduce the robustness of neural networks relying solely on HSV-based representations. To address this, we propose an alternative visualization that augments flow features with additional cues. Specifically, we stack flow magnitudes across three channels, defining the ReynoldsFlow representation as $\boldsymbol{F}_R^n = [|\boldsymbol{v}_o^n|, |\boldsymbol{v}_c^n|, f^n]$, while omitting directional information. In this scheme, the red and green channels encode the magnitudes of the OF and CF, respectively, while the blue channel preserves the current frame's intensity $f^n$, enhancing spatial detail and contrast. This design improves flow clarity and robustness to visual ambiguity, particularly for tiny or fast-moving objects. As visualized in Figure 1, $\boldsymbol{F}_R^n$ yields sharper features across diverse datasets.

## 4 Experimental Results

To evaluate the proposed ReynoldsFlow representation, we conducted experiments on three computer vision tasks: (1) pose estimation on GolfDB (McNally et al., 2019), (2) action recognition on HMDB51 (Kuehne et al., 2011) and UCF101 (Soomro et al., 2012), and (3) object detection on Anti-UAV (Jiang et al., 2021), ARD100 (Guo et al., 2025), and UAVDB (Chen, 2024). We compared ReynoldsFlow with 13 approaches, including 1) original RGB and 2) grayscale videos, classical OF methods, 3) Horn-Schunck (Horn & Schunck, 1981), 4) dense Lucas-Kanade (Lucas & Kanade, 1981), 5) Farneback (Farnebäck, 2003), 6) Brox (Brox et al., 2004), 7) TV-L1 (Zach et al., 2007), 8) DeepFlow (Weinzaepfel et al., 2013), 9) PCAFlow (Wulff & Black, 2015), and 10) DIS (Kroeger et al., 2016), as well as deep learning-based methods such as 11) RPKNet (Morimitsu et al., 2024), 12) SEA-RAFT (Wang et al., 2024), and 13) DPFlow (Morimitsu et al., 2025). OF methods were selected for their training-free nature, aligning with the design philosophy of ReynoldsFlow. Learning-based methods, while capable of higher performance with domain-specific tuning, often require ground truth for training or fine-tuning, which is unavailable in most datasets. Therefore, we used the official pretrained models with the best-reported performance for each method. All flow representations are visualized in the HSV color space using the same visualization scheme, as described in (Liu et al., 2009), where motion magnitude and direction are encoded through hue and saturation, respectively. For the ReynoldsFlow representation, defined as $\boldsymbol{F}_R^n = [|\boldsymbol{v}_o^n|, |\boldsymbol{v}_c^n|, f^n]$, we visualize only the magnitude components and omit directional information, as discussed earlier. All experiments reported in Section 4 were conducted on a high-performance computing system (Meade et al., 2017) equipped with an NVIDIA H100 GPU with 80 GB of memory.

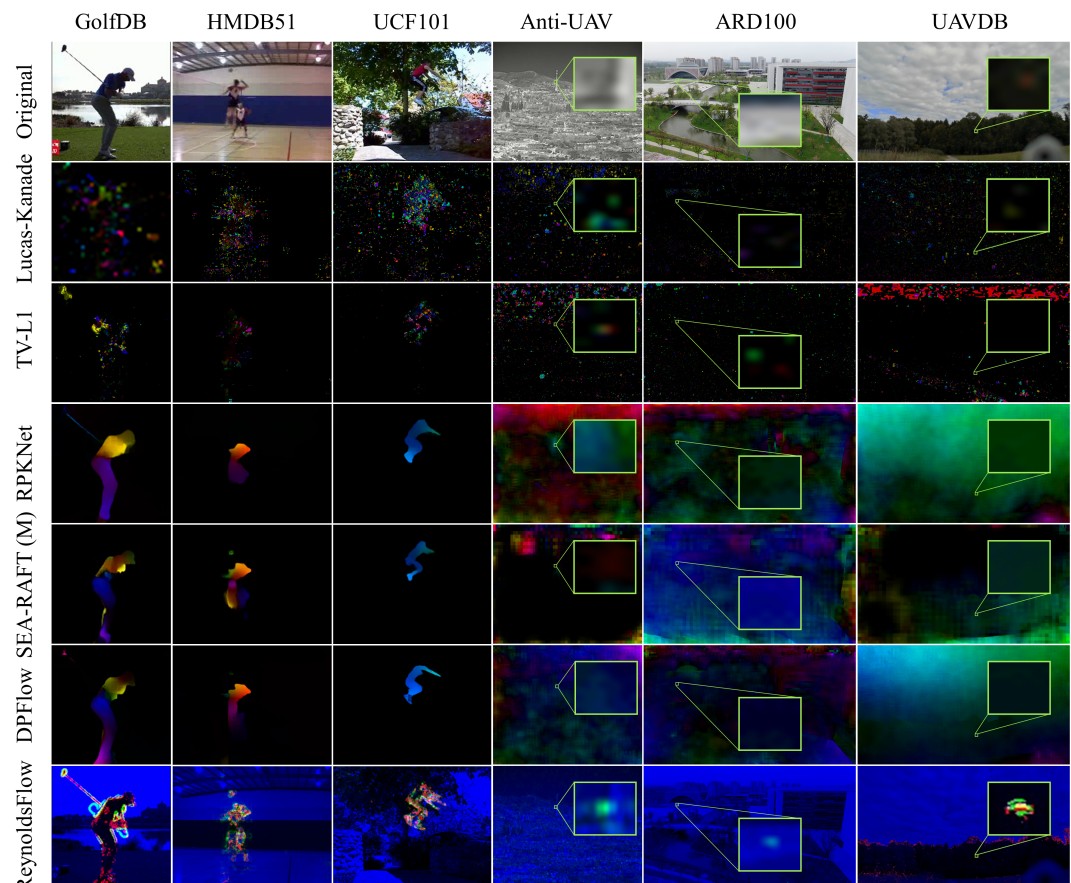

Figure 1: Top row (left to right): example frames from GolfDB (McNally et al., 2019), HMDB51 (Kuehne et al., 2011), UCF101 (Soomro et al., 2012), Anti-UAV (Jiang et al., 2021), ARD100 (Guo et al., 2025), and UAVDB (Chen, 2024). Middle rows: OF visualizations from Lucas-Kanade (Lucas & Kanade, 1981), TV-L1 (Zach et al., 2007), RPKNet (Morimitsu et al., 2024), SEA-RAFT (Wang et al., 2024), and DPFlow (Morimitsu et al., 2025), shown in HSV color encoding. Bottom row: ReynoldsFlow representations for each scene.

## 4.1 POSE ESTIMATION ON GOLFDB

We evaluate the proposed method on the pose estimation task using the GolfDB dataset, which contains 1,400 videos at a resolution of 160×160 pixels. Each video features a subject who occupies nearly the entire frame. The objective is to accurately identify specific poses within the golf swing sequence, which is divided into eight distinct events: Address (A), Toe-up (TU), Mid-backswing (MB), Top (T), Mid-downswing (MD), Impact (I), Mid-follow-through (MFT), and Finish (F). The dataset includes face-on and down-the-line views, offering diverse perspectives to capture the nuanced transitions between each event.

For implementation, we adopt SwingNet (McNally et al., 2019), a model designed for pose estimation in golf swing videos. We maintain the original training configuration, including a sequence length of 64, 10 frozen layers, a batch size of 22, 2,000 iterations, and six workers. We use the pretrained MobileNetV2 model provided by (Sandler et al., 2018) and then fine-tune it on the GolfDB dataset. Performance is evaluated using the Percentage of Correct Events (PCE) metric, with four-fold cross-validation to ensure reliability. For real-time videos sampled at 30 fps, a tolerance of $\delta = 1$ is used. For slow-motion videos, the tolerance is calculated as $\delta = \max(\lfloor \frac{N}{f} \rceil, 1)$, where $N$ is the number of frames from Address to Impact, $f$ is the sampling frequency, and $\lfloor x \rceil$ denotes rounding $x$ to the nearest integer. As shown in Table 1, our input achieves the highest PCE of 0.812.

Table 1: Effect of flow estimation on downstream task performance across GolfDB (McNally et al., 2019), HMDB51 (Kuehne et al., 2011), UCF101 (Soomro et al., 2012), Anti-UAV (Jiang et al., 2021), ARD100 (Guo et al., 2025), and UAVDB (Chen, 2024).

| Methods | GolfDB | HMDB51 | UCF101 | Anti-UAV | | ARD100 | | UAVDB | |
|---|---|---|---|---|---|---|---|---|---|
| | PCE ↑ | Accuracy ↑ | Accuracy ↑ | $AP_{50}^{test}$ ↑ | $AP_{50-95}^{test}$ ↑ | $AP_{50}^{test}$ ↑ | $AP_{50-95}^{test}$ ↑ | $AP_{50}^{test}$ ↑ | $AP_{50-95}^{test}$ ↑ |
| RGB | 0.705 | 0.372 | 0.698 | – | | 0.554 | 0.304 | 0.811 | 0.518 |
| Grayscale / Infrared | 0.698 | 0.328 | 0.614 | 0.781 | 0.418 | 0.376 | 0.167 | 0.660 | 0.281 |
| Horn-Schunck (Horn & Schunck, 1981) | 0.707 | 0.165 | 0.259 | 0.217 | 0.080 | 0.074 | 0.016 | 0.104 | 0.021 |
| Lucas-Kanade (Lucas & Kanade, 1981) | 0.692 | 0.214 | 0.338 | 0.280 | 0.114 | 0.237 | 0.141 | 0.500 | 0.200 |
| Farneback (Farnebäck, 2003) | 0.717 | 0.248 | 0.383 | 0.500 | 0.246 | 0.182 | 0.103 | 0.258 | 0.145 |
| Brox (Brox et al., 2004) | 0.708 | 0.260 | 0.328 | 0.379 | 0.168 | 0.122 | 0.089 | 0.244 | 0.110 |
| TV-L1 (Zach et al., 2007) | 0.810 | 0.284 | 0.537 | 0.600 | 0.278 | 0.227 | 0.127 | 0.779 | 0.409 |
| DeepFlow (Weinzaepfel et al., 2013) | 0.706 | 0.237 | 0.419 | 0.344 | 0.143 | 0.138 | 0.063 | 0.154 | 0.058 |
| PCAFlow (Wulff & Black, 2015) | 0.765 | 0.296 | 0.424 | 0.580 | 0.286 | 0.128 | 0.041 | 0.547 | 0.332 |
| DIS (Kroeger et al., 2016) | 0.772 | 0.207 | 0.562 | 0.347 | 0.144 | 0.102 | 0.018 | 0.151 | 0.057 |
| RPKNet (Morimitsu et al., 2024) | 0.801 | 0.395 | **0.734** | 0.316 | 0.214 | 0.088 | 0.014 | 0.110 | 0.039 |
| SEA-RAFT (M) (Wang et al., 2024) | 0.782 | **0.419** | 0.722 | 0.357 | 0.188 | 0.089 | 0.048 | 0.486 | 0.243 |
| DPFlow (Morimitsu et al., 2025) | 0.786 | 0.411 | 0.705 | 0.427 | 0.265 | 0.072 | 0.016 | 0.270 | 0.101 |
| ReynoldsFlow (Ours) | **0.812** | 0.402 | 0.714 | **0.792** | **0.446** | **0.602** | **0.326** | **0.895** | **0.547** |

## 4.2 ACTION RECOGNITION ON HMDB51 AND UCF101

Beyond pose estimation, we also evaluate action recognition performance on two widely used benchmarks: HMDB51 (Kuehne et al., 2011), which contains 6,766 clips spanning 51 action categories with resolutions ranging from 176×240 to 592×240 pixels, and UCF101 (Soomro et al., 2012), which includes 13,320 clips at 320×240 resolution across 101 categories.

For implementation, we follow the self-supervised video representation learning framework of (Jenni et al., 2020), adopting their C3D-based architecture for action recognition with different flow inputs. We retain the original training configuration, including a batch size of 6, 100 epochs for pre-training, and 75 epochs for fine-tuning. Following standard practice, we report classification accuracy on both datasets using two-fold cross-validation for reliability. As shown in Table 1, although ReynoldsFlow does not achieve the highest accuracy on these benchmarks, it consistently improves over the RGB baseline and performs comparably to learning-based flow methods.

## 4.3 OBJECT DETECTION ON UAV DATASETS

We further evaluate the proposed method on object detection tasks using three distinct UAV datasets: Anti-UAV, ARD100, and UAVDB. Anti-UAV consists of infrared video frames with resolution ranging from 512×512 to 640×512 captured by a moving camera. ARD100 comprises high-resolution RGB videos with resolution 1920×1080 featuring UAVs in diverse environments, with footage captured from Phantom-type drones. UAVDB is also composed of high-resolution RGB frames with resolution ranging from 1920×1080 to 3840×2160 recorded by a static ground camera. All three datasets focus on single-class UAV annotations and include targets at various scales, reflecting real-world, challenging scenarios where object size varies with distance from the camera. Particularly, we selected 4,800 training, 1,600 validation, and 1,600 test images from 223 Anti-UAV videos. For ARD100, which originally contains over 200,000 images, we sampled about one-tenth from each video, resulting in 12,000 training, 4,000 validation, and 4,000 test images from 100 videos. For the UAVDB dataset, we directly used the official split provided by the authors, which includes 10,763 training, 2,720 validation, and 4,578 test images.

In the implementation, we first compute flow estimations using the aforementioned methods, then apply YOLOv11n (Jocher & Qiu, 2024) as the object detector. All models were trained using eight workers, an input resolution of 640×640, a batch size of 64, and for over 100 epochs. Mosaic augmentation was applied throughout training except for the final ten epochs, during which it was disabled to stabilize convergence. We employed transfer learning by initializing from official YOLOv11n pre-trained weights and fine-tuning on the Anti-UAV, ARD100, and UAVDB datasets to incorporate prior knowledge. For evaluation, we report $AP_{50}$ and $AP_{50-95}$ on both validation and test sets. As demonstrated in Table 1, YOLOv11n with our input achieves the highest performance across all datasets, highlighting its effectiveness.

Beyond Table 1, we compare ReynoldsFlow with spatiotemporal embedding models such as TransVisDrone (Sangam et al., 2022), which are tailored for UAV detection. Although these models

Table 2: Runtime comparison (seconds) of OF algorithms on CPU or GPU for UAVDB per image.

| Algorithms | OpenCV Packages | Runtime (s) ↓ |
|---|---|---|
| Horn-Schunck (Horn & Schunck, 1981) | – | 1.951 |
| Lucas-Kanade (Lucas & Kanade, 1981) | cuda_DensePyrLKOpticalFlow | **0.013** |
| Farneback (Farnebäck, 2003) | cuda_FarnebackOpticalFlow | 0.031 |
| Brox (Brox et al., 2004) | cuda_BroxOpticalFlow | 0.093 |
| TV-L1 (Zach et al., 2007) | cuda_OpticalFlowDual_TVL1 | 3.165 |
| DeepFlow (Weinzaepfel et al., 2013) | createOptFlow_DeepFlow | 2.521 |
| PCAFlow (Wulff & Black, 2015) | createOptFlow_PCAFlow | 0.403 |
| DIS (Kroeger et al., 2016) | DISOpticalFlow_create | 0.046 |
| RPKNet (Morimitsu et al., 2024) | – | 1.568 |
| SEA-RAFT (M) (Wang et al., 2024) | – | 1.416 |
| DPFlow (Morimitsu et al., 2025) | – | 1.372 |
| ReynoldsFlow (Ours) | – | 0.019 |

Table 3: Ablation study of ReynoldsFlow representations on downstream task performance.

| Representations | GolfDB | HMDB51 | UCF101 | Anti-UAV | | ARD100 | | UAVDB | |
|---|---|---|---|---|---|---|---|---|---|
| | PCE ↑ | Accuracy ↑ | Accuracy ↑ | $AP_{50}^{test}$ ↑ | $AP_{50-95}^{test}$ ↑ | $AP_{50}^{test}$ ↑ | $AP_{50-95}^{test}$ ↑ | $AP_{50}^{test}$ ↑ | $AP_{50-95}^{test}$ ↑ |
| HSV (Liu et al., 2009) | 0.804 | 0.382 | 0.597 | 0.646 | 0.320 | 0.417 | 0.211 | 0.500 | 0.288 |
| $[\|v_o^n\|, \; - \;, f^n]$ | 0.788 | 0.375 | 0.684 | 0.765 | 0.407 | 0.478 | 0.262 | 0.803 | 0.471 |
| $[\|v_c^n\|, \|v_o^n\|, f^n]$ | 0.791 | 0.367 | 0.699 | 0.784 | 0.386 | 0.509 | 0.287 | 0.831 | 0.492 |
| $[\|v_o^n\|, \|v_c^n\|, f^n]$ | **0.812** | **0.402** | **0.714** | **0.792** | **0.446** | **0.602** | **0.326** | **0.895** | **0.547** |

achieve comparable accuracy, they require over five times longer training and more than six times slower inference. In contrast, ReynoldsFlow adds only negligible overhead relative to model inference, and a detailed runtime comparison is provided in Section 4.4. Moreover, spatiotemporal models demand task-specific preprocessing and retraining to adapt to new applications (e.g., pose estimation), whereas ReynoldsFlow is plug-and-play: the representation can be directly computed from raw video data and seamlessly integrated into downstream models.

## 4.4 ANALYSIS AND DISCUSSION

ReynoldsFlow demonstrates clear benefits in accuracy, efficiency, and representation design. First, while large objects naturally benefit from accurate motion direction cues in video analysis and thus show predictable performance gains across different flow features, the advantage of ReynoldsFlow becomes particularly evident when dealing with tiny or nearly invisible targets. In these cases, directional information is less informative, but motion magnitude remains crucial, allowing Reynolds-Flow to maintain high accuracy where both classical and deep learning methods fail.

Second, ReynoldsFlow achieves strong runtime efficiency. On UAVDB, using an Intel Core i7-12650H CPU and an NVIDIA RTX 4050 GPU, it runs at 0.019 s per image, competitive with classical methods such as Farneback (0.031 s) and DIS (0.046 s), and substantially faster than more complex methods like Brox, TV-L1, and DeepFlow (0.09–3.2 s), and other learning-based approaches (around 1.5 s per image). A complete runtime comparison is provided in Table 2, demonstrating that ReynoldsFlow is well-suited for real-time and embedded applications.

Third, our ablation studies reveal three key insights: (1) **Visualization format:** The ReynoldsFlow representation (without directional information) substantially improves downstream video understanding performance over classical HSV color space visualization (with directional information), as shown by comparing the first and last rows of Table 3. (2) **CF magnitude contribution:** Incorporating the CF magnitude $|v_c^n|$ consistently enhances the representation, evident from the comparison between the second and last rows in Table 3. (3) **Channel-ordering combinations:** Six combinations of stacked visualizations were evaluated, with the top two performances reported in the third and last rows in Table 3. Notably, placing the flow magnitude in the green channel outperforms

placing the current frame's intensity there, which aligns with the RGGB Bayer filter pattern: the human eye is more sensitive to green light and perceives brightness more acutely than color detail.

These results highlight the benefits of incorporating the CF component into ReynoldsFlow and effectively combining representation channels. Overall, the experiments demonstrate that ReynoldsFlow provides robust video representations, runtime efficiency for resource-constrained deployment, and clear performance gains from incorporating CF.

## 5 CONCLUSION

We presented ReynoldsFlow, a physics-inspired video representation that captures both divergence-free and curl-free motion components in a principled, interpretable, and lightweight manner, grounded in the Helmholtz decomposition and Reynolds transport theorem. ReynoldsFlow delivers robust motion representations, improving over the RGB baseline on datasets with large objects and offering clear advantages for tiny or nearly invisible targets. It is plug-and-play, requiring no retraining or task-specific preprocessing. Ablation studies show that incorporating the CF magnitude and using stacked visualizations are crucial for improving performance across multiple datasets, highlighting both interpretability and practical utility. Importantly, potential applications of Reynolds-Flow include broader multi-task video learning scenarios and extensions to handle complex environmental variations, such as dynamic illumination and structural deformations. Moreover, due to its lightweight and efficient design, ReynoldsFlow is particularly well-suited for real-time deployment on resource-constrained platforms.

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
