# OpenReview forum: "ReynoldsFlow: Spatiotemporal Flow Representations for Video Learning"
_ICLR.cc/2026/Conference — Submitted to ICLR 2026_

### Official Review · Reviewer_CWTf · 2025-10-27

**Soundness:** 3
**Presentation:** 1
**Contribution:** 3
**Rating:** 4
**Confidence:** 3

**Summary:**

This paper proposes ReynoldsFlow, a physics-driven spatiotemporal flow representation derived from the Helmholtz decomposition and Reynolds transport theorem (RTT). Unlike conventional optical flow, it explicitly models both divergence-free and curl-free motion components, yielding representations robust to illumination and scale variations. The method is unsupervised, training-free, and computationally lightweight, serving as a plug-and-play motion prior for downstream tasks. When applied to pose estimation and UAV object detection, ReynoldsFlow consistently outperforms classical and deep optical-flow baselines, demonstrating strong generalization and efficiency.

**Strengths:**

S1. Technically novel and theoretically solid: ReynoldsFlow generalizes conventional optical flow by jointly modeling divergence-free and curl-free components, providing a theoretically consistent and physically grounded formulation that rigorously supplements existing flow representations.

S2. Efficiency: The method is training-free and lightweight, making it practical for real-time and embedded settings.

S3. Broad applicability: ReynoldsFlow shows consistent gains across pose estimation and UAV detection, demonstrating task-agnostic adaptability as a plug-and-play flow representation.

**Weaknesses:**

W1. Lack of controlled robustness analysis: Although the authors claim resilience to illumination and scale changes, no controlled experiments isolate or quantify these effects. For example, perturbation tests with varying lighting or object scaling would substantiate these claims.


W2. Limited validation on temporal understanding tasks: Most experiments focus on image-level tasks (pose and object detection). Evaluation on video-level understanding tasks (e.g., UAV action recognition, object tracking) would better demonstrate generalization to broader spatiotemporal understanding.

W3. Loss of directional information: ReynoldsFlow representation omits explicit motion direction in its representation, retaining only the magnitudes of divergence-free and curl-free components. While this simplifies visualization, direction cues are important for tasks like action recognition or tracking. A justification or ablation on this design choice would strengthen the argument.

W4. Insufficient figures & tables: It would be better if the authors provided figures and tables about results in Sec.4.3. to clearly demonstrate the effectiveness of the proposed methods.

**Questions:**

Q1. The paper highlights that ReynoldsFlow performs particularly well on tiny objects; could the authors clarify how its representation enables more accurate motion capture for small-scale targets?

Q2. Are there observed failure cases where ReynoldsFlow underperforms?

Overall, I found the paper interesting, with clear technical novelty and strong theoretical grounding. However, a well-controlled analysis comparing ReynoldsFlow with existing flow estimation methods is essential (see W1). Without a detailed and controlled analysis of the proposed method, I have no choice but to give a borderline reject, though the rating could be raised if these weaknesses are adequately addressed. Additional improvements on W2–W4 would also be beneficial.

---

> ### Author Response · Authors · 2025-11-18
> **Responses to the Official Review by Reviewer CWTf**
>
> **Response to Weakness 1**
>
> Thank you for bringing this up. The video datasets we use may already contain natural variations in lighting, which challenge classical optical flow methods due to their reliance on brightness constancy. Similarly, UAV videos inherently include scale changes as the camera or target moves closer or farther. While these factors are implicitly present in our current experiments, we agree that a more controlled robustness analysis with explicit perturbations would further substantiate these claims and plan to explore this in future work.
>
>
> **Response to Weakness 2**
>
> Thank you for pointing this out. In the revised rebuttal PDF, we have added additional experiments on general video understanding tasks, including action recognition benchmarks on HMDB51 and UCF101. These benchmarks cover the broader temporal and spatiotemporal understanding tasks highlighted by the reviewer, helping to further demonstrate the generalization and applicability of ReynoldsFlow beyond image-level tasks.
>
>
> **Response to Weakness 3**
>
> Thank you for the valuable suggestion. Accurate motion direction can indeed benefit pose estimation and action recognition tasks, as shown in Table 1 of the rebuttal PDF. However, under varying illumination, obtaining reliable directional information (HSV) becomes challenging. This issue is particularly pronounced for tiny objects, where illumination changes make it difficult to distinguish the target, Figure 1 illustrates that classical HSV visualization often fails to localize UAV targets. As a result, using direction as a feature for object detection or recognition may not be robust. Also, the ablation study now covers three aspects:
>
> 1. **Visualization format:** comparing classical HSV flow (with direction) versus stacked representation (magnitude only) (first row vs. last row in Table 3).
> 2. **CF magnitude contribution:** evaluating performance with and without incorporating the CF component into the representation (second row vs. last row in Table 3).
> 3. **Channel-ordering combinations:** analyzing the top two stacked configurations (third row vs. last row in Table 3).
>
> Notably, the first aspect directly addresses the role of directional information. Comparing HSV (with direction) to magnitude-only representations (Table 3, first vs. last row) provides quantitative insight into the impact of including or omitting direction in the representation.
>
>
> **Response to Weakness 4**
>
> Thank you for mentioning this. We have uploaded a revised rebuttal PDF with updated figures and detailed tables to better illustrate the effectiveness of ReynoldsFlow:
>
> 1. Adding an action recognition benchmarks (HMDB51, UCF101) comparison,
> 2. Updating Figure 1 and Table 1 with new results,
> 3. Adding Table 2 and Table 3 for runtime comparisons and detailed ablation studies.
>
> These updates provide more visual and quantitative evidence of the method's performance across a broader range of tasks.
>
>
> **Response to Question 1**
>
> Thank you for bringing this up. We believe ReynoldsFlow works particularly well on tiny objects due to two main factors:
>
> 1. **Directional information:** Motion direction is often unreliable for tiny objects, as they are highly sensitive to illumination changes. By combining CF and OF magnitudes, ReynoldsFlow provides a robust motion representation that captures subtle displacements even when directional cues are weak. This makes it especially effective for detecting and understanding small, fast-moving targets, where OF methods often struggle.
> 2. **ReynoldsFlow representation:** Rather than relying on HSV-to-RGB transformations, which are highly nonlinear and can introduce perceptual inconsistencies, especially in low-texture regions, complex illumination, or dynamic motion scenarios, ReynoldsFlow directly stacks the features into three channels. This maintains a more consistent and informative representation, improving motion capture for small-scale targets.
>
>
> **Response to Question 2**
>
> Thank you for the question. Across our experiments, ReynoldsFlow consistently improves downstream video understanding performance compared to the RGB baseline. That said, it does not always surpass the best-performing OF methods on all datasets, particularly for scenarios with large, smooth-moving objects where classical or learning-based flows already capture motion very accurately. Nevertheless, ReynoldsFlow achieves these improvements with substantially lower computational cost, offering a favorable trade-off between efficiency and performance.

---

### Official Review · Reviewer_hLGr · 2025-11-01

**Soundness:** 3
**Presentation:** 3
**Contribution:** 3
**Rating:** 4
**Confidence:** 2

**Summary:**

This paper proposes a physics-inspired video spatiotemporal flow representation method called ReynoldsFlow, which directly derives spatiotemporal features with physical interpretability from video data. The method adopts an unsupervised and training-free design, constructing texture-preserving and dynamic-aware feature representations by integrating flow field magnitude with frame intensity. It performs exceptionally well in tasks such as small object detection and pose estimation.

**Strengths:**

1. Lightweight plug-and-play utility: ReynoldsFlow directly computes from raw video frames without training process and large-scale annotated data, with computational cost comparable to traditional optical flow methods. Its three-channel representation (optical flow magnitude, complementary flow magnitude, frame intensity) can be seamlessly integrated into existing detection and pose estimation models, without task-specific preprocessing or model reconstruction, offering strong engineering practicality and compatibility.

2. The paper breaks through the limitations of traditional optical flow that rely on the brightness constancy assumption, introducing fluid dynamics into video spatiotemporal feature modeling, and first proposes ReynoldsFlow which simultaneously includes traditional optical flow (divergence-free component) and complementary flow (curl-free component). This design covers information beyond motion, such as illumination changes and non-rigid deformations, from a principled perspective, solving the robustness issues of traditional optical flow in complex scenarios, and the physical modeling endows features with stronger interpretability.

**Weaknesses:**

1. The generalization capability of the task coverage is insufficient: the experimental validation focuses on two specific tasks, UAV target detection and golf swing pose estimation, and lacks performance evaluation on more general video understanding tasks (such as action recognition, video semantic segmentation). The current results are insufficient to fully demonstrate the generalization ability of ReynoldsFlow, especially its performance in scenarios with non-rigid targets and dense motion trajectories is still unclear. I believe that the dataset's performance on more video tasks will help in verifying the model's capabilities.

2. The data typesetting in Table 1 is chaotic and the data is missing, which affects the readability of the results.

**Questions:**

Please refer to the weaknesses.

---

> ### Author Response · Authors · 2025-11-18
> **Responses to the Official Review by Reviewer hLGr**
>
> **Response to Weakness 1**
>
> Thank you for the valuable suggestion. In the revised rebuttal PDF, we have added additional experiments on general video understanding tasks, including action recognition benchmarks on HMDB51 and UCF101. These benchmarks fall within the broader video understanding scope highlighted by the reviewer. Regarding non-rigid and dense motion scenarios, the tested datasets already contain such characteristics. For example, UAV videos naturally include non-rigid motion patterns (e.g., targets moving closer or farther from the camera), and both the action recognition datasets and the Anti-UAV dataset contain dense and complex motion due to non-stationary cameras. These experiments provide evidence that ReynoldsFlow remains applicable and stable under such conditions. We expect that further exploration on additional video tasks, such as video semantic segmentation and more diverse non-rigid motion datasets, would strengthen the demonstration of generalization.
>
> **Response to Weakness 2**
>
> Thank you for pointing this out. We have corrected the formatting and updated Table 1 to improve readability and ensure all data is clearly presented. Additionally, Table 2 and Table 3 now include more detailed results for runtime comparisons and the comprehensive ablation study.
>
> **Response to Questions**
>
> Thank you for your questions. We have addressed each identified weakness in detail above.

---

### Official Review · Reviewer_NJ3u · 2025-11-06

**Soundness:** 4
**Presentation:** 4
**Contribution:** 3
**Rating:** 4
**Confidence:** 3

**Summary:**

This paper introduces a novel spatiotemporal flow representation method named ReynoldsFlow for video representation learning. Unlike traditional 3D convolutional modules or optical flow-based networks, ReynoldsFlow is inspired by physics, utilizing Helmholtz decomposition and the Reynolds transport theorem to extract theoretically-grounded spatiotemporal features directly from video data. This method is unique in its ability to capture both divergence-free and curl-free components, making it robust to photometric variations (such as illumination and scale changes) while preserving the intrinsic structure of the video. Experiments demonstrate that ReynoldsFlow is a lightweight and flexible representation, combining frame intensity and flow magnitude, and significantly enhances performance in tasks like tiny object detection.

**Strengths:**

1. The paper introduces fundamental principles from fluid dynamics (Helmholtz decomposition, Reynolds transport theorem) into deep learning, providing a novel and physically-constrained framework for spatiotemporal feature extraction, which possesses high originality in the video learning domain.
2. Results show that the method can "substantially enhance tiny object detection," a recognized challenge in video analysis, highlighting its practical value in high-resolution or complex scenes.
3.The paper mentions ReynoldsFlow is "lightweight and adaptable," suggesting it likely avoids the massive computational overhead associated with 3D convolutions or complex optical flow networks.

**Weaknesses:**

1. Details of Main Experiments: Are the OF method results in Table 1 reproduced by the authors? What representation method is used for optical flow?
2. GolfDB Performance: ReynoldsFlow may not outperform OF on GolfDB, as classical TV-L1 achieves 0.81 and learning-based methods typically improve with fine-tuning. When v_n^c​ is removed, the method relies on OF information yet underperforms the best OF results.
3. Ablation Study: The ablation study appears limited. Additional combinations of F_R^n should be tested to better understand each component's contribution.
4. Complementary Claims: The paper claims complementarity to OF but only evaluates datasets where CF excels. Testing on general datasets would strengthen this claim, as true complementarity should not degrade performance on any task.

**Questions:**

Please see Weaknesses 1-3

---

> ### Author Response · Authors · 2025-11-18
> **Responses to the Official Review by Reviewer NJ3u**
>
> **Response to Weakness 1**
>
> Thank you for the question.
>
> 1. Yes, all results including classical OF methods and learning-based baselines, were reproduced by us on each dataset.
>
> 2. For OF methods, flow fields are visualized in the HSV color space following the same visualization through~\citep{liu2009beyond}, where hue encodes motion direction and saturation encodes magnitude. For our ReynoldsFlow representation, $F_R^n = [|v_o^n|, |v_c^n|, f^n]$, we visualize only the magnitude components, omitting directional information, as discussed in the Section 3.3 ReynoldsFlow Representation.
>
>
> **Response to Weakness 2**
>
> Thank you for bringing this up. Exactly, for datasets with relatively large and smooth-moving objects, such as GolfDB, HMDB51, and UCF101, existing optical flow algorithms, including classical TV-L1 and learning-based methods, already capture sufficient motion information to benefit downstream video understanding tasks, as shown in Table 1. In these cases, ReynoldsFlow also improves the RGB baseline and achieves comparable performance to the best OF methods. Moreover, ReynoldsFlow offers lower computational cost, as demonstrated in Table 2. Our primary focus, therefore, is on more challenging scenarios involving tiny, fast-moving, and arbitrarily moving objects, where conventional OF methods often fail to extract reliable motion cues. In such cases, ReynoldsFlow provides both robust representations and efficiency advantages.
>
>
> **Response to Weakness 3**
>
> Thank you for the suggestion. We have added a more comprehensive ablation study in the revised rebuttal PDF. The ablation study now covers three aspects:
>
> 1. **Visualization format:** comparing classical HSV flow (with direction) versus stacked representation (magnitude only) (first row vs. last row in Table 3).
> 2. **CF magnitude contribution:** evaluating performance with and without incorporating the CF component into the representation (second row vs. last row in Table 3).
> 3. **Channel-ordering combinations:** analyzing the top two stacked configurations (third row vs. last row in Table 3).
>
> These ablation studies provide a clearer understanding of how each aspect of $F_R^n$ contributes to overall performance.
>
>
> **Response to Weakness 4**
>
> Thank you for pointing this out. In the updated rebuttal PDF, we added evaluations on general action recognition benchmarks to assess the effect of ReynoldsFlow. While ReynoldsFlow does not surpass the best OF methods on these datasets, it consistently improves over the RGB baseline, demonstrating that incorporating the CF term does not degrade performance in video understanding tasks. These results support our claim of complementarity: ReynoldsFlow provides clear benefits in challenging tiny-object scenarios while maintaining non-negative performance impact on broader, general-purpose datasets.
>
>
> **Response to Questions**
>
> Thank you for your questions. We have addressed each identified weakness in detail above.

---

### Author Response · Authors · 2025-11-18
**Response Overview and Summary of Revisions**

Thank you for the time and effort spent reviewing our work. Below, we provide detailed responses to each weakness and question raised by the reviewers. We have also uploaded a revised rebuttal PDF. The main updates include:
1. Adding an action recognition benchmarks (HMDB51, UCF101) comparison,
2. Updating Figure 1 and Table 1 with new results,
3. Adding Table 2 and Table 3 for runtime comparisons and detailed ablation studies.

Please feel free to leave any additional comments if further clarification is needed. We greatly appreciate all the constructive suggestions and feedback.

---

### Meta-Review · Area_Chair_BisD · 2026-01-08

**Summary:**

This paper proposes ReynoldsFlow, a physics-inspired spatiotemporal flow representation that leverages Helmholtz decomposition and Reynolds transport theorem to extract divergence-free and curl-free motion components. It provides a lightweight, training-free, and interpretable feature descriptor that enhances video understanding tasks, particularly for tiny object detection under photometric variations.
1. Reviewers NJ3u and hLGr pointed out that the experimental validation was limited to specific tasks like UAV detection and pose estimation, lacking evaluation on more general video understanding tasks such as action recognition.
2. Reviewer NJ3u and CWTf noted that the method might not outperform existing optical flow (OF) methods on datasets with large, smooth-moving objects (e.g., GolfDB) and questioned the loss of directional information in the representation.
3. Reviewer CWTf raised concerns regarding the lack of controlled robustness analysis to isolate the effects of illumination and scale changes, which were claimed as strengths of the paper.
4. Reviewer hLGr and CWTf mentioned presentation issues, including chaotic data typesetting in Table 1 and a lack of sufficient figures/tables to demonstrate effectiveness in certain sections.

Considering the lack of controlled experiments to prove the core physical robustness claims and the marginal performance gains on general video tasks, I tend to reject this paper.

**Reviewer Concerns:**

1. The authors added action recognition benchmarks (HMDB51, UCF101) in the rebuttal. I believe this partially addresses the concern, though the performance does not surpass SOTA learning-based OF methods on these general tasks.
2. The authors provided a comprehensive ablation study on visualization formats and channel-ordering. While they justified that magnitude-only representations are more robust for tiny objects, the inherent loss of direction remains a fundamental limitation for general motion modeling.
3. The authors acknowledged that a controlled robustness analysis with explicit perturbations was missing and planned to explore it in future work. I believe this core claim remains empirically unsubstantiated in the current version.
4. The authors updated Figure 1 and Table 1 and added runtime comparisons. This effectively resolved the readability and presentation concerns.

**Reviewer Scores:**

- Reviewer NJ3u: 4 -> 4. While action recognition results were added, the method still underperforms compared to classical TV-L1 on GolfDB when certain components are removed, and the complementarity claim remains weak on general datasets.
- Reviewer hLGr: 4 -> 6. Concerns 1 and 4 were well addressed through the inclusion of new benchmarks and improved table formatting.
- Reviewer CWTf: 4 -> 4. Concern 3 remains unresolved as no controlled perturbation tests were provided to quantify the claimed robustness to illumination and scale changes.

---

### Decision · Program_Chairs · 2026-01-26

Reject